# An Integrated View of Virus-Triggered Cellular Plasticity Using Boolean Networks

**DOI:** 10.3390/cells10112863

**Published:** 2021-10-24

**Authors:** Jenny Paola Alfaro-García, María Camila Granados-Alzate, Miguel Vicente-Manzanares, Juan Carlos Gallego-Gómez

**Affiliations:** 1Molecular and Translation Medicine Group, Faculty of Medicine, University of Antioquia, Medellin 050010, Colombia; jenny.alfaro@udea.edu.co (J.P.A.-G.); camila.granados@udea.edu.co (M.C.G.-A.); 2Molecular Mechanisms Program, Centro de Investigación del Cáncer, Instituto de Biología Molecular y Celular del Cáncer, Consejo Superior de Investigaciones Científicas (CSIC)-University of Salamanca, 37007 Salamanca, Spain

**Keywords:** cellular plasticity, epithelial–mesenchymal transition, systems biology, Boolean, viral infection

## Abstract

Virus-related mortality and morbidity are due to cell/tissue damage caused by replicative pressure and resource exhaustion, e.g., HBV or HIV; exaggerated immune responses, e.g., SARS-CoV-2; and cancer, e.g., EBV or HPV. In this context, oncogenic and other types of viruses drive genetic and epigenetic changes that expand the tumorigenic program, including modifications to the ability of cancer cells to migrate. The best-characterized group of changes is collectively known as the epithelial–mesenchymal transition, or EMT. This is a complex phenomenon classically described using biochemistry, cell biology and genetics. However, these methods require enormous, often slow, efforts to identify and validate novel therapeutic targets. Systems biology can complement and accelerate discoveries in this field. One example of such an approach is Boolean networks, which make complex biological problems tractable by modeling data (“nodes”) connected by logical operators. Here, we focus on virus-induced cellular plasticity and cell reprogramming in mammals, and how Boolean networks could provide novel insights into the ability of some viruses to trigger uncontrolled cell proliferation and EMT, two key hallmarks of cancer.

## 1. Introduction

In 1985, Helen Blau endowed the term “cellular plasticity” with its current meaning, that is, the ability of specific cells, under certain conditions, to turn into other cell lineages. Using cell fusion-based methods, she established the relationship between genotype and phenotype during cellular regeneration and tissue homeostasis [1]. The gist of her paradigm-shifting discovery was that the differentiated state of a cell is subject to continuous regulation and revision, determining the identity of the cell within its surrounding tissue [2]. In this context, plasticity refers to the activation of “dormant” genes that change the identity of the cell in response to microenvironment stimuli, e.g., viral infection or mechanical injury [3,4].

Cellular plasticity rapidly derived into the discovery of stem cells (SCs) and the recognition of their therapeutic potential [4]. Cloning of a whole mammal (Dolly the sheep) using cellular reprogramming was an early milestone in the field [5,6]. This achievement was regarded as irrefutable evidence that differentiated cells could be reprogrammed into other lineages, including progenitors capable of initiating an entire morphogenetic program culminating in the de novo formation of an entire complex organism. Roughly 10 years after the initial description of Dolly, terminally differentiated mouse cells were driven back to full stem potential by expression of selected transcription factors [7]. These reprogrammed cells were termed “inducible pluripotent stem cells”, or iPS. These milestones ignited a revolution in the field of regenerative medicine, as stem cells from different origins (hematopoietic, mesenchymal, embryonic or inducible) were used to obtain differentiated cells from multiple lineages [8]. In parallel, stemness became an important concept in cancer research, with the discovery of heterogeneous cancer cell populations with stem cell-like properties. These cells were termed cancer stem cells, or CSCs [9]. CSCs are crucial for primary tumors to develop resistance to therapy and relapse, as well as metastasis. CSCs also provided a potential explanation for decades-old observations describing that most forms of cancer were morphologically based on the dedifferentiation of the original tissues [10]. These observations linked cancer with the reactivation of dormant morphogenetic pathways that become inactivated after terminal differentiation [11].

A classic categorization of plasticity includes three major forms: (1) Dedifferentiation, in which a differentiated cell reverts to a SC of the same lineage; (2) Trans-differentiation, in which a differentiated cell is converted into another lineage without passing through a pluripotent cell state; (3) Trans-determination, in which a cell changes its lineage from a SC or progenitor cell to a closely related cell type [12].

Although their numbers are low, SCs do exist in healthy adult tissues, indicating that they play physiological roles in homeostasis [13,14]. Based on their ability to differentiate into tissue-forming cells, their main function is to renew and regenerate tissues during normal aging, and also in response to injury. The homeostasis of these cell populations also involves dedifferentiation or trans-differentiation of fully, or partially, differentiated cells [15,16,17]. In some of these tissues, e.g., intestinal crypts, SC adopt a quiescent phenotype, differentiating in response to specific functional needs [18].

Several families of viruses have evolved to hijack different pathways and circuits involved in cell plasticity to ultimately serve the infectious program. As such, they may alter the homeostatic landscape of the host’s tissues, in some cases triggering plasticity programs that may have deleterious effects in the integrity of the tissue, for example, driving cancer. What follows is a discussion on the different mechanisms of virus-induced plasticity, their meaning in the context of the propagation of the infectious program, and how Boolean networks (a form of systems biology analysis) can unravel the intrinsic complexity generated by the intersection of apparently different cellular programs.

## 2. Cell Plasticity and Viral Infection: A key Role for EMT in Virus-Induced Cell ReProgramming

Many studies have revealed how viral infection can trigger cell reprogramming, leading, in many cases, to the development of cancer. The best-characterized program of cancer-related cell reprogramming is the epithelial–mesenchymal transition (EMT). EMT is also the most common mode of cell reprogramming among viruses, underlying the oncogenic capacity of many virus strains. EMT has been reviewed in multiple contexts [19,20,21]. Briefly, EMT promotes the dedifferentiation of epithelial cells into mesenchymal cells, which are more primitive and motile. In many cases, this is linked to the ability of virus proteins to deactivate tumor suppressor genes, or behave as oncogenes [22,23,24], driving tumorigenesis in infected cells. Virus-induced mesenchymal cells acquire the ability to manipulate the microenvironment of the affected cells to promote proliferation and migration. These changes are associated with modifications of the repertoire of transcription factors expressed by infected cells. In turn, this modulates the profile of cytoskeletal, adhesive, and metabolic cellular components, altering the shape and behavior of the cell in response to these changes [25,26]. Here, we only discuss virus families that are able to drive EMT, and the mechanisms involved.

Different types of viruses induce EMT through different mechanisms (shown in Figure 1). One example is the Human Papilloma Virus (HPV; Papillomaviridae). More than 130 subtypes have been identified [27]. HPV infects stratified basal epithelia, which normally appears in anatomical regions (skin, cervix) that contain many more cell populations, including undifferentiated cells, progenitors, immune cells, etc. HPV induces cancer transformation. Two viral proteins, E6 and E7, target p53 and Rb, respectively [28]. E6 and E7 also alter the function of Rb-like protein DREAM (dimerization partner, RB-like, E2F4, and MuyB) and c-myc [29]. E6 and E7 also trigger EMT in infected cells by activating mesenchymal-driving transcription factors, e.g., ZEB1, SLUG and TWIST1—inducing the expression of mesenchymal genes, e.g., N-cadherin, vimentin and fibronectin [30]. Simultaneously, E6 and E7 down-regulate E-cadherin expression [31,32].

Epstein–Barr virus (EBV; Herpesviridae), which infects B cells [33] and epithelial cells [34], also induces EMT. Its genome encodes two protein products termed latent membrane protein (LMP) 1 and 2A. Although not essential for infection [35], these proteins trigger EMT. LMP1 promotes EMT by inducing the expression of TWIST1 [36] and SNAIL [37] transcription factors, inducing an E-to-N-cadherin switch [38]. LMP2A also promotes EMT in nasopharyngeal carcinoma (NPC) cells, increasing their invasion and migration capabilities [39] in an mTOR1-dependent manner [40]. Other EBV protein products, e.g., EBNA-1 and EBNA3C, also promote EMT [41]. The specific signaling routes involved in EBV-induced EMT are not precisely defined, although they involve several components typically involved in EMT, including the JAK/STAT and PI3/AKT pathways [42]. Additional mechanisms involve the regulation of specific miRNA. EBV expresses Bart9, which is homologous to cellular microRNAs miR-200 and miR-141. These two miRNA are involved in EBV-mediated downregulation of E-cadherin, as demonstrated by the fact that their deletion using siRNA increases the levels of E-cadherin in gastric cancer cells [43].

Human cytomegalovirus (HCMV; beta-herpesvirus type 5) [44] also infects epithelial cells, promoting EMT by a mechanism that involves TGF-β1 activation in a model of renal tubular cells [45]. However, gene set enrichment analysis (GSEA) of breast cancer and glioma stem cells revealed the repression of a mesenchymal phenotype correlative with E-cadherin expression [46], which contradicts the previous study and suggests that the plasticity effect of this virus may be cell type- or stage-dependent. On the other hand, Kaposi sarcoma virus (KSHV) infects endothelial cells, triggering endothelial–mesenchymal transition (EndMT) through the Notch signaling pathway [47]. Interestingly, KSHV also infects B cells, and EMT markers are also important in the transformation of these cells, even if they do not define a specific morphological change in these cells [48].

Hepatitis-causing viruses also trigger EMT. HBV (Orthohepadnavirus) has four overlapping ORFs [49]. The X ORF encodes protein X (HBx) [50], which modulates the function of several transcription factors by favoring the formation of the transcriptional initiation complex. HBx also promotes EMT by stabilizing SNAIL, positively regulating TWIST and STAT3 and activating c-Src [51,52,53]. HBx also represses E-cadherin expression [54].

HCV (hepacivirus) expresses several nonstructural proteins [55]. Among these, NS3, NS4B and NS5A induce EMT by diverse mechanisms. NS3 promotes EMT through an epigenetic mechanism that boosts TGF-β signaling [56]. NS4B increases the expression of the transcription factor SNAIL, triggering EMT in liver cancer [57], while NS5A up-regulates TWIST2 [58] and activates β-catenin-dependent signaling [59]. In addition to NS proteins, other viral products also trigger EMT. The HCV core protein (HCVc) inhibits Smad3 activation, promoting EMT. HCVc also bypasses the Ras/PI3K signaling route to activate ERK, JNK and p38, promoting a signaling signature typical of EMT, which includes the stabilization of HIF-1α [60].

Human immunodeficiency virus 1 (HIV-1) induces EMT through the Hedgehog pathway in podocytes, becoming a critical feature of kidney damage in a mouse model of HIV infection [61]. HIV-1 gp120 and tat proteins also induce EMT in polarized, squamous, oral, cervical, and genital epithelia [62].

Human respiratory syncytial virus (RSV) belongs to the Pneumoviridae family [63] and causes airway hyper-responsiveness in infants [64]. RSV induces EMT in a nodal-dependent manner [65]. Human rhinoviruses (RV) also trigger EMT in vitro in bronchial cell lines [66].

The COVID-19 pandemic makes it necessary to address whether SARS-CoV-2 also induces EMT/EndMT. Analysis of post-mortem samples has suggested that SARS-CoV-2 may trigger EndMT in the lungs [67]. A more recent report has indicated that SARS-CoV-2 induces EMT in the lung by targeting the transcription factor ZEB1 [68]. This could be very important for its pathogenesis, since ACE2, which is the main cellular receptor of the virus in various cell types, is downregulated during EMT [68]. EndMT may also be critical for the pathogenesis of COVID-19-dependent lung fibrosis [69], as crosstalk between epithelial and endothelial cells mediates infection-driven injury to human alveolar capillaries [70]. In addition, infected cells become p38-positive downstream of the mesenchymal transcription factor SNAIL [71]. p38 triggers cytoskeletal rearrangement during EMT [72]. A different study using post-mortem samples revealed increased levels of neutrophil extracellular traps (NET), which induce EMT in neighboring cells [73]. The increased presence of NETs positively correlates with the severity of COVID-19 and the elevated expression of EMT markers [74]. Importantly, a recent study showed that sera obtained from COVID-19 patients induce EMT in cancer cells, although the specific mechanism and whether SARS-CoV-2 makes patients more vulnerable to more aggressive forms of cancer remain to be determined [75].

At this point, the evolutive advantage that viruses obtain by inducing EMT/EndMT in host cells is unclear. What is clear is that virus-driven EMT events correlate with the severity of the cancer triggered by viral infection. For example, a preprint study has indicated that Coxsackievirus B3 (CVB3)-induced EndMT is countered by bone morphogenetic protein 7 (BMP7), suggesting that EndMT, which correlates with poor prognosis, can be targeted to improve the patient’s odds of survival [76]. It is also important to highlight that EMT/EndMT can affect the bioavailability of the virus receptors, indicating that cells in different stages may have different susceptibilities to infection. As mentioned above, EMT represses ACE2, which is the main receptor of SARS-CoV-2 [68]. Likewise, the measles virus is not capable of infecting epithelial cells undergoing EMT, likely due to the fact that the virus receptor is negatively regulated during EMT [77]. A key question in the field is whether EMT is a side effect of viral infection, or if there is an evolutive purpose from the point of view of the virus. This is a fascinating question as the field moves forward.

## 3. Understanding EMT from a Systems Biology Point of View

In biology, systems approaches often refer to the use of big data to study complex biological phenomena. “Big data” includes analysis of –omics approaches, e.g., genomic, epigenomic, and proteomic and/or simulation-based data arrays. A major advantage of these approaches is that they are, by default, unbiased; hence, they often lead to highly novel, “out-of-the-box” insights. Perhaps the biggest challenge of this approach is post-discovery validation in biased systems, which are better characterized, but provide a more boxed-in approach. Canonical systems biology can be characterized as an iterative process in which large-scale analysis of a limited set of perturbations in a well-characterized biological system allows development of predictive models that can be further refined by additional experimentation.

Based on the type of data analyzed, there are two major approaches to the analysis of large-scale data. One is bottom-up, in which input data emanate from different pre-existing resources, including genomics, biochemical and metabolic organism-specific databases. These data are used to rebuild a draft, which is then curated manually prior to generating mathematical models that incorporate refining algorithms applied iteratively to improve the predictions of the model. The other approach is top-down, which involves de novo data generation, which is analyzed using bioinformatics tools that enable the construction of interactive pathways based on the statistical significance of the experimental results [78].

EMT provides a fascinating example of application of systems biology approaches, which have provided important insights. Specifically, EMT has been studied using systems biology approaches in the context of asthma. A bottom-up approach using existing rodent datasets revealed a high degree of heterogeneity even among genetically identical individuals, and a specific signature of up-regulated genes in asthma [79]. Another study used high-content proteomics and metabolomics data to classify asthma patients according to protein expression in the respiratory system, opening the door to the discovery of novel biomarkers with potential applications in diagnosis and therapy [80]. Other datasets revealed that increased levels of adipokines and ROS-related protein products are predictors of severity [81]. Along the same lines, systems approaches have revealed crucial molecular signatures related to EMT in the context of cancer, including the existence of three separable states: pre-EMT, metastable EMT and epigenetically fixed EMT [82]. The use of BNs to study cancer has enabled advances in different directions. For example, a recent study has used BNs to perform tumor staging and assess the risk of metastasis in patients with triple-negative breast cancer. This approach identified hybrid epithelial/mesenchymal phenotypes by mapping gene expression data into the states of a Boolean network model of the epithelial–mesenchymal pathway [83]. This study built on a previous study from the same group that used transcriptomic data to create a BN-based topographic map of EMT, showing the existence of multiple intermediate, metastable states—determining a continuum of states between stable (purely epithelial and purely mesenchymal) phenotypes [84]. Another study also evaluated the dynamic stability of these intermediate states using BNs [85]. BN have also been used to validate and extend the observations using “wet” approaches, e.g., 3-D models in decellularized matrices. In such a system, application of TGF-beta induced cell invasion, which was predicted as well by use of an EMT-driven BN [86]. EMT-driven BNs can also be used to interrogate the effect of combinatorial therapies to treat cancer, as shown for hepatoma [87]. A recent review has proposed that cancer signaling-driven BNs will be useful for investigating the prognosis of and therapeutic responses to different interventions, contributing to the improvement of the decision-making process in oncology practice [88].

## 4. Boolean Networks in Systems Biology-Driven Analysis of EMT

Boolean networks (BN) are dynamic models that connect multiple variables that behave in a binary manner. This means that the variables are assumed to be –false or –true, and no intermediate states contribute to the stability or evolution of the model over time. While this is a large assumption, this approximation is valid for many enzymatic reactions and protein–protein interactions, which are governed by sigmoidal equations that define two stable states (0 and 1) connected by a metastable, steep-sloped, state. The system has *n* variables, X = [x_1_, x_2_… x_n_] that represent the components of the system. Boolean operators, AND, OR and NOT, connect these variables [89].

There are different models of BNs according to their static and dynamic properties and their evolution over time. The most commonly used are: deterministic and non-deterministic (referring to the nature of the variables), synchronous and asynchronous (referring to the updating of the BN over time), homogeneous and non-homogeneous, directed and undirected, regular and non-regular [90]. A brief description of the most biologically relevant of these models follows.

Recent application of BNs to biology subdivides variables into three categories: (i) Input nodes (top upstream), which are not regulated by other variables. Typical input nodes in cell biology are extracellular signals that are used to induce external perturbations in the model; (ii) Output nodes (endpoints), which do not regulate other variables; and (iii) Inner nodes, which connect input and output nodes forming a network. In this manner, these variables can be represented as a connections map, in which Boolean operators determine the value of each variable (except input nodes, which are arbitrarily defined) at a given time, t. The possible combinations of –false or –true assignments to all the variables that comprise the network define the evolution of the BN over time. It is important to highlight that the processing steps of BNs are loosely related to real time. In this regard, an intrinsic limitation of BNs is that time behaves as a discrete variable; hence, the state of the BN is updated from t to t + 1 by applying the Boolean operators that connect the different components. The t → t + 1 transition is managed according to several different paradigms: Synchronous, in which every Boolean operator is applied simultaneously to update the BN from t to t + 1. This assumes that all the biological processes represented by Boolean connectors take the same amount of time, which is not frequent in biological processes; Asynchronous, which assumes that one (and only one) variable is updated in the t → t + 1 transition. Although asynchronicity reflects real biological scenarios more faithfully than synchronicity, several major issues remain: one is that this system poorly replicates the actual time scale of cellular functions, in which some processes are very fast (e.g., protein–protein interactions) and some are much slower (e.g., protein synthesis). Asynchronous modeling also disregards the hierarchical nature of some connections over others. For example, a protein cannot become phosphorylated before being synthesized. Finally, the run time of asynchronous modeling can be an important practical limitation [91]. However, software tools that implement logical modeling, e.g., GINSim, enable the operator to solve this problem with the possibility of prioritizing the updating of classes and/or interacting nodes according to convenience [92]. Another choice to overcome these issues is the asynchronous paradigm, which has been modified into stochastic, random order update, general and deterministic models. In stochastic models, each time step is updated using a random permutation in the nodes, while in deterministic models, there is a preselected time unit for each node, which is updated based on the positive multiples of the established unit (either random or imposed) [91,93]. These strategies have been conceived to model different time scales while preserving the deterministic nature of synchronous updating, or to overcome the temporal distortion of asynchronous updating. Asynchronous and stochastic updating allow the incorporation of the concept of multi-stability—that is, the simultaneous presence of multiple space-state attractors, each one endowed with its own level of attraction. Stochastic multi-stability produced by the asynchrony of a BN is very well-suited to describe some biological processes [94,95]. One example is the analysis of p53 in the DNA damage response, which displays two-face dynamics [96]. A recent study used BNs to explain how the p53 pathway determines the fate (apoptosis, autophagy or senescence) of cells transfected with miR-16 [97]. Likewise, a recent study explored multi-stability in EMT using DSGRN (dynamic signatures generated by regulatory networks). The major conclusion of this study was that whereas epithelial and mesenchymal states are the dominant attractors, the system contemplates the existence of bi- or multi-stability [98]. A major extension was the introduction of probabilistic Boolean operators (e.g., the probability of AND is 80%; whereas the probability of OR is 20% for the same connection). However, it is worth noting that, in general, synchronous updating is a better strategy to evaluate the robustness of BN models [99].

BNs were introduced in biology as they elegantly explained genetic regulation [100,101]. In this scenario, expression is considered a binary proposition, where −false = not expressed and −true = expressed [102]. This approximation also enables following of the evolution of specific nodes within the BN [93]. Employing scale-free topology, certain commonalities emerge, e.g., the fact that most nodes are poorly connected to others, whereas some nodes are highly connected (“hubs”). This setup, general to many biological processes, ensures robustness—providing resistance against missing, or defective, nodes. It is important to point out that, whereas BNs work well for systems that can be binarized, as described above, they work poorly in systems in which protein concentration is a crucial element, e.g., the behavior of cytoskeletal components, in which small concentrations of “nucleators” catalyze polymerization and thus control the functionality of the network.

A transition from one time point to the next generates representations termed “state graphs”, which depend on the evolution paradigm used. When applying a synchronous model, state graphs evolve in a single manner, that is, each state has one possible successor alone. Conversely, asynchronous and probabilistic models have complex state graphs in which a node has several possible evolutions depending on whether the node is selected for update, or not. An important consequence of state graphs is that their trajectory defines the overall behavior of the BN over time, revealing states that define stable conditions, that is, states from which the BN does not evolve unless a perturbation is introduced. These states are termed attractors, and represent system equilibrium across different conditions. In synchronous models, attractors are simple, formed by a sequence of states repeated over time. In asynchronous models, complex attractors emerge, defined as overlapping loops formed from the multiple possible successor states at specific nodes. Finally, probabilistic models do not always contain attractors, as some possible updates are intrinsically unstable (low probability).

Attractors generate basins of attraction, which are defined as all the states leading to a given attractor. Thus, the larger the basin, the more likely the attractor is to correspond to a meaningful biological condition. Due to their nature, synchronous models display non-overlapping basins of attraction, whereas asynchronous models display less well-defined basins due to the non-deterministic nature of the model [89].

As they become applied in biology, it is important to note that BN can be built using bottom-up or top-down approaches. The bottom-up approach involves beginning with prior knowledge, or literature mining. Pre-existing data sets and bibliographic information define the nature of the relationship between the variables of the system. Bioinformatics tools are available to assist in this process [89,103,104]. On the other hand, top-down approaches employ novel data sets to construct the BN. This requires the binarization of the data sets, which is easy for genomics and transcriptomics data, as discussed above. However, proteomics data require additional data processing, e.g., establishing arbitrary thresholds of expression. Then, reconstruction algorithms are employed to infer interactions or Boolean relationships between variables. There are multiple methods to create reconstruction algorithms, which depend on the specifics of the problem and the available amount of experimental data. In two excellent reviews, Le Novere [105] and Abou-Jaoudé and co-workers [106] detail different methods and approaches to formalize and validate working models that implement quantitative and logic modeling. They also discuss methodological advances that facilitate the development of cellular networks by assessing the dynamic impact of variations and external inputs, determining the major attractors of large networks and reducing large models. Finally, they discuss examples of models of published biological processes, public databases and options to upgrade BN modelling [105,106].

To our knowledge, no studies have reported the use of BNs to model virus-induced EMT or EndMT. The recent COVID-19 pandemic has caused an outburst of bottom-up models to explain different aspects of the pandemic, from virus transmission [107] to the effect of the pandemic on mental health [108], or the response to specific drugs, e.g., tocilizumab [109]. Examples more pertinent to the discussion here include the use of BNs to model the signaling involved in the infection by influenza virus [110] or the interaction of HIV with T cell signaling [111]. An open-ended literature search and integration in a BN model was recently described to determine some epidemiologic aspects of ZIKV [112].

Regarding the use of BN to specifically study EMT, a recent publication described a synchronous updating BN to model the stability of transitional states during EMT [85]. In this study, the authors selected five variables (extracellular conditions that trigger EMT and four genes critically involved in EMT: SNAIL, miR-34, ZEB, miR-200). By establishing hierarchical and feed-forward and feedback loops among them, the authors determined three states (note the order of variables: Trigger, SNAIL1, miR-34, ZEB1, miR-200): epithelial (0,0,1,0,1), intermediate (1,1,0,1,1) and mesenchymal (1,1,0,1,0). Whereas the epithelial and mesenchymal states are unique, different combinations could represent the intermediate state, e.g., (1,0,0,1,0), (1,1,1,0,1) and others. BN validation was based on wet biology experiments, in which miR-200 or miR-34 expression suppressed mesenchymal states; whereas expression of SNAIL1 and ZEB1 expression suppressed epithelial states. By calculating the size of the basins of attraction for each state, the authors showed that the epithelial phenotype is slightly more stable than the mesenchymal (50% vs. 34% of possible states in each basin), whereas intermediate states are much less likely (only 16% of possible states). It is worth noting that this model works well due to the small number of nodes (five). Larger BNs are due to run into longer computational times and the weight of stochastic events will likely muddy the determination of basin sizes, and thus, the stability of the intermediate states. Despite this limitation, this study articulates, among other things, the concept of EMT reversibility, which is a key feature of this process during development and disease [113,114].

BNs have also been used to determine novel pluripotency genes involved in the reprogramming of SCs in germ lines [115] and to describe EndMT in cardiac cells [116]. In the latter study, the authors built a BN that connected several molecules involved in endothelial cell activation and EndMT. EndMT is important during heart valve development, whereas altered EndMT underlies some heart defects, atherosclerosis and pulmonary hypertension. Using a synchronous BN model in which the authors determine the nodes as morphological conversions (apical–basal to front–back polarization, decreased cell adhesion and repression of endothelial and emergence of mesenchymal markers), the authors determined that EndMT requires increased expression of WNT, NOTCH, FGF or TGF in the extracellular microenvironment, the activation of transcription factors as SNAIL1/2, TWIST1, ZEB1/2, and inhibition of VEGFR2, PECAM1, VE-Cadherin, TIE1, TIE2 and vWF. Conversely, mesenchymal cells express increased amounts of α-SMA, N-cadherin and collagen I/II. The authors also analyzed the robustness of the model by introducing data describing the effect of loss- or gain-of-function mutations affecting selected nodes in the BN. Importantly, only 24 of 58 mutations had no effect on the final outcome as determined by the type of resulting attractors, which indicates that most nodes included in the BN are crucial for the whole process. In the end, state transition graphs connecting the diverse intermediate states and their molecular makeup revealed that not every transition is likely. For example, the transition from tip ECs (which mediate angiogenesis) to mesenchymal cells is very unlikely. It also revealed that EndMT requires loss of VEGFA and no hypoxia, together with the inactivation of FLI1 and GATA2, whereas loss of SNAIL2, TWIST1 or ZEB1/2 prevent EndMT. Perhaps more impressively, the analysis also revealed that mechanical perturbations, e.g., non-laminar shear flow, triggers EndMT, and predicted several genetic/expression alterations that revert EndMT.

## 5. Conclusions and Perspectives

Whereas the potential of systems biology in general, and BN in particular, to unravel the complexity of virus-induced plasticity events is clear, the studies and data discussed here have illustrated some of the limitations of these approaches, which may underlie the fact that the field is still underdeveloped. As a result, several key questions in the field remain unanswered. Perhaps the most pressing question from a conceptual standpoint is to reveal the ultimate goal of different families of viruses to promote EMT in host or bystander cells, particularly as, in some cases, increased plasticity negatively affects the viral cycle [77]. Another goal of the implementation of these techniques to study virus-induced EMT is to realize the potential of determining novel therapeutic targets that counter the carcinogenic transformation brought on by oncogenic viruses. The present review also provides a conceptual framework that justifies the use of BNs as a first approach to understand EMT, as they are useful tools for integrating the genetic, epigenetic and transcriptomic complexity of this process. A major success of these approaches to date has been the determination of the existence of a continuum of intermediate states between the purely epithelial and purely mesenchymal end states. This concept alone suggests the existence of multiple points of convergence and divergence, redundancies and vulnerabilities, which may reveal “choke” points and therapeutic intervention points.

A major challenge emerges in applying these approaches in a real biological setting. Often, data procurers are basic “wet” scientists that possess extensive, evidence-based knowledge of different signaling pathways involved in the processes under study, but they lack the “know-how” regarding the implementation of systems biology approaches. Conversely, systems biology bioinformatics experts often lack basic knowledge regarding regulation and signaling. This is a disadvantage that may lead to incorrect results or delays, as such knowledge enables the rapid curation of the predicted data links and possible outcomes. Ideally, BNs can be used to understand how virus-induced EMT and EndMT occur, directing experimentation and cooperating with the interpretation of biological experiments (Figure 2). However, widespread implementation is often marred by the inability of these two types of experts to interact efficiently.

Major hurdles include the day-to-day design of BNs in a “wet lab” context, and the limited availability of curated data sets. The former is eased by the development of multiple tools available for modelling BNs, including GINSim [92], BooleanNet (R package) [117], PicoSAT solver [118], VisiBool [119], Boolsi [120], Boolesim [121], jSBGN [122] and others. In addition, additional data sets including transcriptomic, genetic and epigenetic data emerge every day, including NIH-curated databases (https://www.cancer.gov/research/resources/search?from=0&toolTypes=datasets_databases, accessed on 17 October 2021), the most prominent of which is TCGA (The Cancer Genome Atlas, https://portal.gdc.cancer.gov/, accessed on 17 October 2021); COSMIC (Catalogue of Somatic Mutations in Cancer, https://cancer.sanger.ac.uk/cosmic, accessed on 17 October 2021) and many others. It is worth noting that many researchers now have the capacity to generate their own datasets through diverse–omics techniques.

In conclusion, this review invites a major implementation of computational science and the use of models like Boolean Networks to study complex events like viral infection and cellular plasticity induced by viruses. Such widespread use of BNs could accelerate our understanding of these complex phenomena and facilitate the search for novel and repurposed antiviral treatments, targeting crucial cellular targets to control virus progression and/or virus-induced cellular transformation.

## Figures and Tables

**Figure 1 cells-10-02863-f001:**
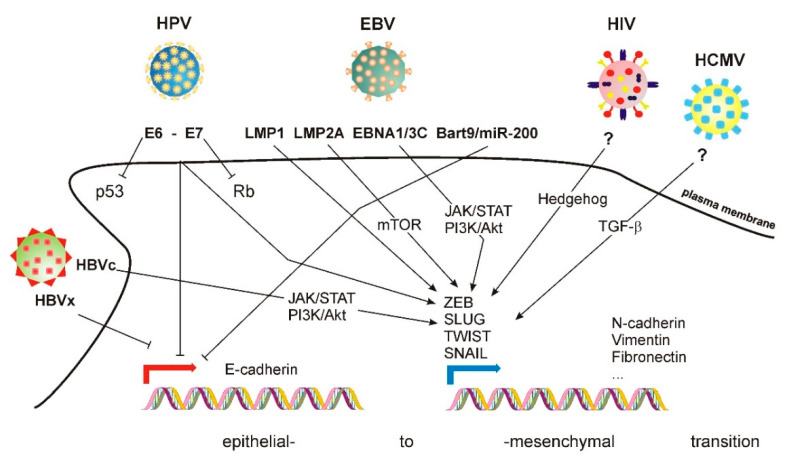
Mechanisms of EMT induced by several families of viruses. See accompanying text for details.

**Figure 2 cells-10-02863-f002:**
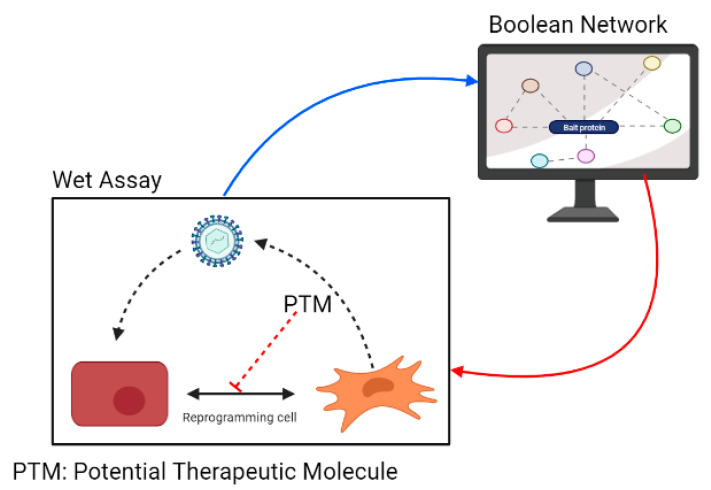
Implementation of BNs to study virus-induced EMT and EndMT and evaluate molecules that can interfere with cellular mechanisms that are activated by viral pathogens.

## Data Availability

Not applicable.

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
