# Peer review of "An Integrated View of Virus-Triggered Cellular Plasticity Using Boolean Networks"

_cells, 2021, doi:10.3390/cells10112863_

Round 1
Reviewer 1 Report
The review appears in general interesting but there major points that should be clarified.
- It is unclear the connection with COVI19. It should be better to explain in a subparagraph which type of virus are shown to be involved in EMT and if there is any connection with COVID19. References are also missing.
- The abstract and the conclusion should be revised considering my comments at point 1.
- BN and EMT is not a new task I recommend to add a paragraph describing the existing literature between the plasticity, EMT and BN in tumors not related to virus.
- In general the main goal of this review is not clear and the conclusions are not completely convincing.
Reviewer 2 Report
Authors review the molecular effects of virus-induced cell plasticity and suggest the use of boolean networks to study these processes. I think their argument is sound, the paper is well written and comprehensive despite being short. However, I have some criticisms that should be addressed before acceptance.
- The processing steps of a boolean network are loosely related to real time. This a problem of different techniques of simulation, not only BN. So, authors should stress that in their text.
- Regarding their statement in lines 253-5, it is possible to prioritize the order of operations that should occur before than others in boolean models, solving that problem. This is available in some modeling tools such as GINSim.
- Another important aspect of the use of the asynchronous update is the emergence of stochastic behavior, specially multistability. There has been a growing recognition that different cell fates (apoptosis or senescence) can emerge from stochasticity due to DNA damage which can only be reproduced by asynchronous BNs. There are many papers in the literature focusing on this problem using continuous and discrete models. Authors should cite some references, for example: Zhang et al. PNAS doi.org/10.1073/pnas.1100600108 / Gupta et al. DNA Repair 10.1016/j.dnarep.2020.102971 and others.
- Authors reveal ignorance on important reviews of BNs that should be cited: Le Novere 10.138/nrg3885 / Abou-Jaude et al. 10.3389/fgene.2016.00094
- Authors should list some available BN tools such as GINSim, BooleanNet and others for the interested reader.
Round 2
Reviewer 1 Report
I think that the authors answered my questions. It can be accepted.
Reviewer 2 Report
I'm satisfied with the changes of the text provided by the authors.